# Synchronization Analysis of a New Four-Dimensional Time-Delay Lorenz System and Its Circuit Experiments

**Zhiyong Cui [1,]\*** , **Dongbo Zhong [2]** and **Xiaohong Qiu [1]**

1   School of Software Engineering, Jiangxi University of Science and Technology, Nanchang 330013, China
2   School of Automation, Nanjing University of Science and Technology, Nanjing 210000, China
\*   Correspondence: zhiyong49@hotmail.com

**Abstract:** Time-delay chaotic systems with multiple positive Lyapunov exponents have been extensively studied in the field of information security. This paper proposes a new four-dimensional time-delay Lorenz system and its chaotic synchronization through the Lyapunov–Krasovskii theory. The sufficient conditions for the stability of the new chaotic system are obtained by the Routh–Hurwitz criterion, and the control parameters are found to have a significant impact on the speed of synchronous convergence. Furthermore, oscillation circuit simulation is essential for contributing the chaotic system to practical applications. Accordingly, the circuits of the chaotic time-delay system and its coupled synchronous control circuit are innovatively designed by Multisim. Experiment results illustrate the behaviors of various attractors in the new time-delay Lorenz system and the effectiveness of the proposed asymptotic synchronous method.

**Keywords:** time-delay system; chaotic synchronization; asymptotic stability; circuit simulation





## 1. Introduction

Chaos is an emerging interdisciplinary field developed with the rapid development of modern science and technology, especially with the emergence and widespread use of computer technology. The dynamic behavior of a chaotic system drastically depends on its initial conditions [1]. A new system is derived in this condition, even if there is a small change in the initial values. The chaotic system appears to be a random behavior, but in reality, this behavior of chaos follows a natural order. Chaotic systems are difficult to control due to their random dynamic behavior, heavy dependence on initial conditions, and pseudo-randomness [2]. Many researchers have extended their fascinating exploration of mathematics and found a variety of new simple systems that can exhibit chaotic states [3–7]. For example, logistic [8], Hénon's [9], Chen [10], and Lü [11] systems. In these maps, initial conditions and unstable fixed points of the chaotic attractors are located nearby. Moreover, J. Sprott found in his book some new systems, simpler models of chaotic systems that are more concise in terms of the values of some system parameters, special symmetries, and dynamics [7]. Chaotic phenomena exist in the scientific fields of geology, biology, and social science. The phenomenon has ergodicity, and is nonperiodic, such as noise-like characteristics [12], making the issue of chaotic synchronous control issue become a study in the field of nonlinear science.

Since the synchronization of chaos was first discovered by Pechora and Carroll in 1990 [13], there has been considerable interest in this research for its potential usage in secure communications, radars, chemical processes, and engineering applications. These depend on the designing of a strategic approach that enables the dynamic behavior of the master system to control the dynamic behavior of the slave system [14–16]. Various control methods have been proposed for synchronization of chaos, such as sliding synchronous mode control [17], the sliding-mode control approach [18], adaptive control methods [19], adaptive observer-based synchronous strategies [20], linear feedback controllers with

unknown parameters [21], predictive methods of a hyperchaotic system [22], backstepping synchronization based on the equivalent transfer function method [23], and a time-delay control [24]. In [15,25], the authors analyzed the exponential synchronization for time-delay perturbation chaotic systems with multiple positive Lyapunov exponents and the applications to secure communications.

In practical engineering applications, such as communications, automation, biology, and chemistry, uncertainties due to time delays cannot be avoided, leading to multiple variations in the dynamic behavior of the actual system and increasing the uncertainty and complexity of the system model. In such time-delayed systems, the dynamic behavior of the system is more difficult to control and apply in practice. Hence the problem of stability analysis and control of systems with time delays has become a considerable and topical area for researchers in this field. Therefore, researchers have conducted various experiments on uncertainty [26,27] and systems with time delay [28–32]. In [27], the nonlinear control hyperchaotic Rossler system with uncertain parameters is proposed. In [26], an adaptive synchronous strategy is introduced to control uncertain dynamical system time delays based on parameter identification. Ref. [11] presents a new state feedback control method to generate a hyperchaotic lv attractor. A delayed model with a third-degree exponential polynomial is proposed and used to control testosterone secretion [32]. Different synchronous mechanisms in nonidentical time-delay maps are studied [31]. By a fuzzy fractional-order neural network, an adaptive synchronization for a class of fractional-order time-delay uncertain chaotic systems is yielded [30]. In [29], analytical estimation is applied to synchronization in coupled time-delay systems. An iterative learning control strategy has been proposed to synchronize two chaotic systems with a free couple and free time delay [28]. On the other hand, this has given rise to a variety of chaotic signal synchronous control mechanisms and programs, with only one positive Lyapunov exponent of low-dimensional chaotic system complexity, resulting in a less secure confidential system [33]. However, the chaotic time-delay system possesses an infinite-dimensional state space and abundant dynamic behaviors in nonlinear fields. For example, Lyapunov exponents for complex systems with delayed feedback are studied [34]. Wang et al. proposed a novel control method for heterogeneous uncertain chaotic systems with time delay, in which a robust framework for synchronous error estimation is presented [35]. Moreover, it has been used for secure communications through multiple heterogeneous chaotic systems in engineering applications.

Using synchronized chaotic systems for real-world applications largely relies on circuit implementations. Thus, the chaotic system of circuit simulation has widespread concern from researchers in this field [36,37]. Ref. [37] proposed a new four-dimensional chaotic system of a circuit simulation program to map out the trajectory of the high-dimensional chaotic attractor. In [38], the author presented numerical simulations that assume a crucial role in their investigation due to chaotic systems being so challenging to understand analytically. Numerous researchers are confident in the relevance of system modeling and circuit simulation, developing new models with components that can present unpredictable, chaotic behavior [39–41]. Models of systems and circuits are applicable in different fields and initiated the exploration of other chaotic oscillation circuits [42–44]. Sprott designed a simple chaotic jerk circuit in 2011 [40], which includes only a few electronic components, such as one inductor, two capacitors, one adaptive resistor, and a nonlinear resistor. His works illustrated similar chaotic behavior to the trajectories around the equilibriums of the system. Lorenz was the first to propose continuous-time chaotic systems [45]. After that, various discrete-time chaotic hyperchaotic systems have been proposed in the literature [46–50], while typical ordinary differential equations are used to represent most types. Chaotic behavior involves exploiting the time-delay paths in autonomous Boolean networks [51,52]. Furthermore, chaos was initially implemented with electronic logic circuits (logic gates and field-programmable gate arrays FPGAs) [53,54]. In [55], dynamical behaviors are analyzed using two new asymmetric chaos circuit oscillators with no dependence on incommensurate time delays.

One of the items that can be studied for uncertain time-delay systems is the broadband, nonperiodic and unpredictable time-delay system. On the other hand, time-delay chaos systems can exhibit more complex hyper-chaotic behaviors. Many positive Lyapunov exponents could be generated even if a low-dimensional time-delay system is used. Those properties offered by chaotic time-delay signals are suitable for signals used in electronic engineering and information processing in particular secure communication systems. Furthermore, compared to the typical low-dimensional chaotic system well studied, the chaotic time-delay system, which has more complex hyper-chaotic behaviors, is rarely reported. Significantly, the chaotic time-delay circuit hinders the practical application of the randomness time-delay systems in secure communication. Accordingly, this research aims to develop a novel chaotic time-delay system and asymptotical synchronous conditions based on the Lyapunov–Krasovskii functional theory.

Moreover, the realization of a high-dimensional delay chaotic synchronization system was introduced, which is not limited to the dimension of the system while generating more than one positive Lyapunov exponent, both high reliability and preventing the cost of the improvement and synchronous difficulty. Specifically, a four-dimensional time-delay Lorenz system oscillation circuit was investigated to ensure the performance of the attractor. In addition, the time delay in the chaotic attractor has abundant dynamic characteristics, which laid the foundation for enhancing data security levels in the secure telecommunications field. The experiment results show that the design is practical, feasible and can be applied in many time-delay dynamic-related scopes.

This paper is organized as follows: In Section 2, the basic chaotic systems used in this paper are presented. In addition, the time-delay disturbance term was introduced in the mathematical model of classical four-dimensional systems to construct a new effective chaotic system. The novelty synchronous controller between the master system and slave system with a self-time-delay disturbance is analyzed and simulated in Section 3. Section 4 innovatively presents a straightforward methodology to design an integrated circuit, particularly, the lag time oscillator module. Time series, phase space reconstruction, and Poincare maps validate the chaotic behaviors of the designed circuit. In Section 5, the nonlinear coupled global asymptotic synchronization of the time-delay Lorenz system and its circuit simulation is the novelty proposed. Finally, conclusions are presented in Section 6.

## 2. Numerical Simulation

The famous American meteorologist, Professor Lorenz E.N, first described Lorenz systems in the 1963 [45]. This paper adopts a three-dimensional Lorenz chaotic system as the following set of third-order differential equations.

$$\begin{cases} \dot{x}(t) = a(y(t) - x(t)) \\ \dot{y}(t) = -x(t)z(t) + cy(t) \\ \dot{z}(t) = x(t)y(t) - bz(t) \end{cases} \tag{1}$$

where $x(t)$, $y(t)$ and $z(t)$ are the state variables of the system, $a$, $b$ and $c$ are the control parameters of the system.

When the system parameters $a = 36$, $b = 3$, and $c = 20$, the system is in a chaotic state, and the trajectory of the system's solution in phase space is obtained. Figure 1a shows the three-dimensional phase diagram on the $x$-$y$-$z$, and Figure 1b–d shows the phase diagram on two planes. Figure 1b–d show the trajectory of chaotic attractors on a two-dimensional plane.

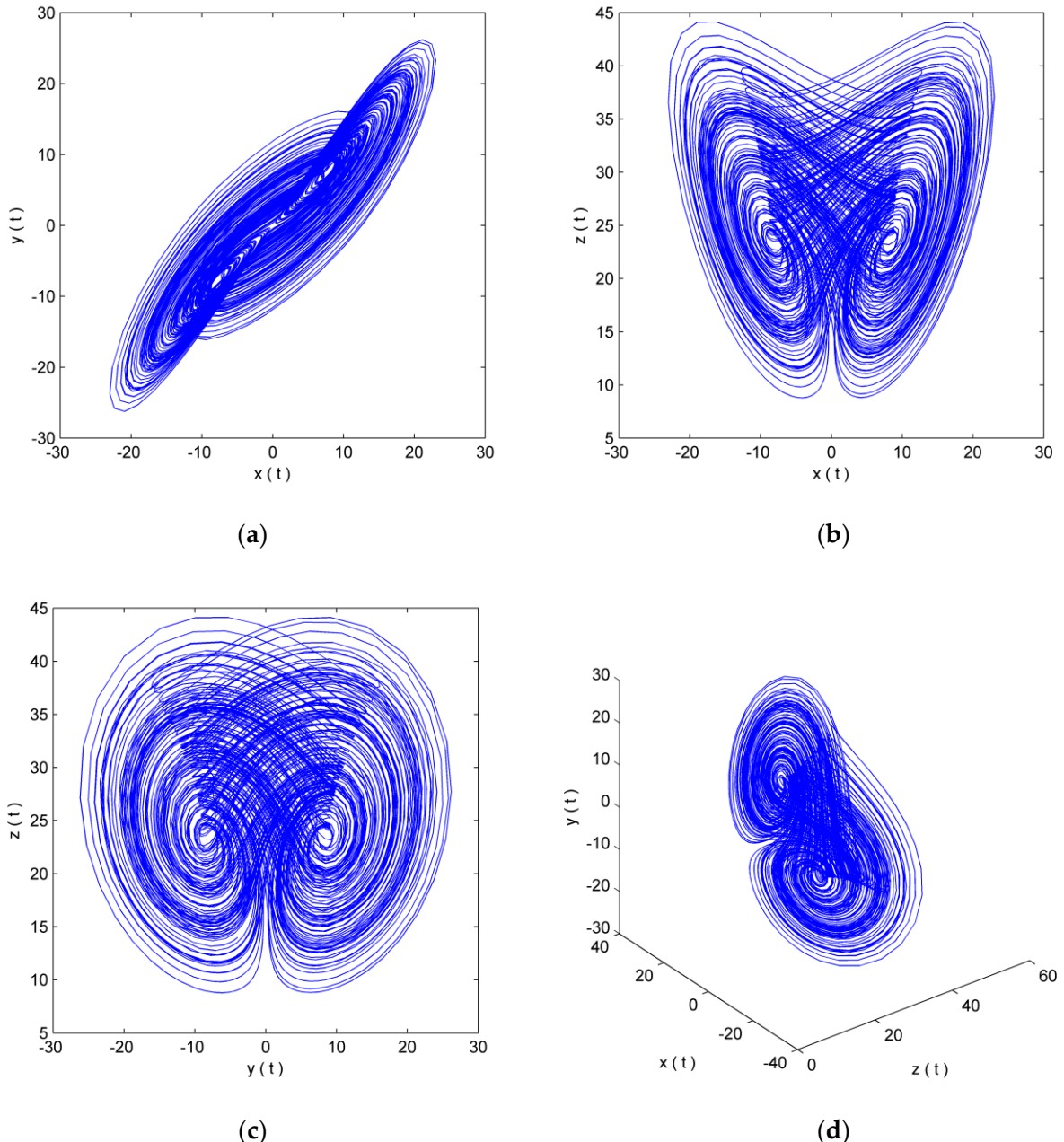

**Figure 1.** Phase diagram of a three-dimensional chaotic attractor. (**a**) *x-y* plane phase diagram; (**b**) *x-z* plane phase diagram; (**c**) *x-z* plane phase diagram; (**d**) three-dimensional diagram.

Chaotic motion is locally unstable, while the whole is stable. The initial value of a chaotic system is affected by minor disturbances. The distance between the points on the orbit and the corresponding points in the original orbit increases exponentially over a period of time. Under the influence of the boundedness of the chaotic motion, the distance between the two points oscillates randomly again.

Further, the initial sensitivity experiment was carried out with the initial values chosen as $(-3, -4, 14)$ and the step size chosen as $h = 0.01$. The calculated curve of $x$ with time $t$ is shown in Figure 2a, Figure 2b shows the waveform $x_1(t)$ after adding a perturbation $10^{-8}$ to the initial function $x(t)$, and Figure 2c shows the difference between them.

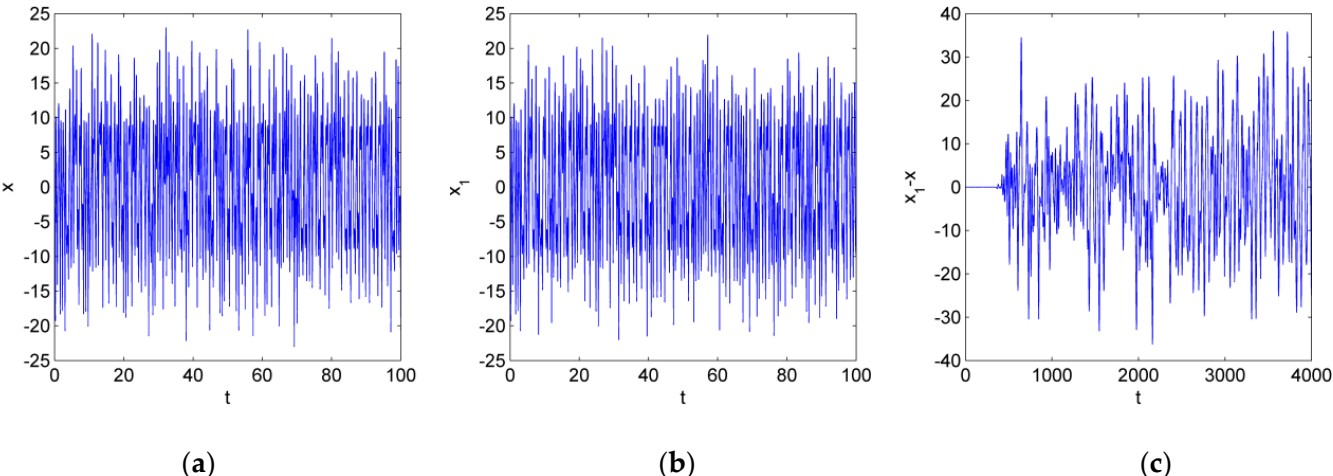

**Figure 2.** Time course diagram. (**a**) Original $x$; (**b**) after adding perturbation $x_1$; (**c**) difference plot $x_1$-$x$.

Let $\dot{x} = \dot{y} = \dot{z} = 0$ gives

$$
\begin{cases}
a(y(t) - x(t)) = 0 \\
-x(t)z(t) + cy(t) = 0 \\
x(t)y(t) - bz(t) = 0
\end{cases}
\tag{2}
$$

calculate the equilibrium points of system (1) and get the result that if $bc \leq 0$, system (1) has only one equilibrium point $S_0(0,0,0)$, and if $bc > 0$, system (1) has three equilibrium points $S_-(-\sqrt{bc}, -\sqrt{bc}, c), S_0(0,0,0), S_+(\sqrt{bc}, \sqrt{bc}, c)$.

Linearizing the system (1) on the equilibrium point $S_0$ to obtain three eigenvalues: $\lambda_1 = -a, \lambda_2 = c, \lambda_3 = -b$. If $c > 0$, the origin is a saddle point in three-dimensional space; if $c < 0$, the origin is the only equilibrium point.

The following is an analysis of the non-zero balance points of the system (1) so that the two non-zero balance points $S_-, S_+$ are $S(x_s, y_s, z_s)$. Linearize the system (1) on two non-zero equilibrium points, and calculate the Jacobian matrix

$$
\begin{vmatrix}
-a - \lambda & a & 0 \\
-z_s & c - \lambda & -x_s \\
y_s & x_s & -b - \lambda
\end{vmatrix} = 0
\tag{3}
$$

The characteristic polynomial is obtained

$$
\lambda^3 + (a + b - c)\lambda^2 + (ab - ac - bc + x_s^2 + az_s)\lambda + (ax_s^2 + abz_s + ax_sy_s - abc) = 0
\tag{4}
$$

substitute the values $S_-(-\sqrt{bc}, -\sqrt{bc}, c), S_+(\sqrt{bc}, \sqrt{bc}, c)$ of the non-zero equilibrium point

$$
\lambda^3 + (a + b - c)\lambda^2 + ab\lambda + 2abc = 0
\tag{5}
$$

Assume the coefficients in the characteristic polynomial are $a_0 = 1, a_1 = a + b - c, a_2 = ab, a_3 = 2abc$. Since the sign of the real part of the eigenvalues of the derived linear system can be determined by using the Routh Hurwitz criterion. It can be inferred whether the corresponding nonlinear system is stable.

According to the Routh–Hurwitz criterion, there are

$$
\begin{aligned}
\Delta_0 &= a_0 = 1 > 0 \\
\Delta_1 &= a_1 = a + b - c \\
\Delta_2 &= \begin{vmatrix} a_1 & a_0 \\ a_3 & a_2 \end{vmatrix} = \begin{vmatrix} a + b - c & 1 \\ 2abc & ab \end{vmatrix} = a^2b + ab^2 - c - 2abc \\
\Delta_3 &= a_3 = 2abc
\end{aligned}
\tag{6}
$$

The equilibrium point $S$ of the system (1) is stable only if it meets $\Delta_1 = a + b - c > 0$, $\Delta_2 = a^2b + ab^2 - c - 2abc > 0$, $\Delta_3 = 2abc > 0$, and Equation (4) has three negative real roots.

The equilibrium point $S$ of the system (1) is unstable when $\Delta_1 = a + b - c < 0$ or $\Delta_2 = a^2b + ab^2 - c - 2abc < 0$ or $\Delta_3 = 2abc < 0$, and Equation (4) has a negative real part root and a pair of imaginary conjugate roots of the positive real part. Both equilibrium points are saddle focus points in three-dimensional space.

System (1) is a dissipative system with a dispersion of

$$\nabla V = \frac{\partial \dot{x}}{\partial x} + \frac{\partial \dot{y}}{\partial y} + \frac{\partial \dot{z}}{\partial z} + \frac{\partial \dot{w}}{\partial w} = -(a - c + b) = -19$$
$$V(t) = V(0)e^{-19}$$

(7)

Since $(a - c + b) > 0$, the system (1) is always dissipative and converges in the exponential form of $\dot{V} = e^{-(a-c+b)}$. Further, an initial volume $V(0)$ converges to a volume element $V(0)e^{-19}$ at time $t$. Therefore, when $t \to \infty$, each volume element containing the system's trajectory shrinks to zero at an exponential rate $-(a - c + b)$. Therefore, all system trajectories will eventually be limited to a set of points with zero volume, and its asymptotic behavior will be fixed on an attractor.

On the other hand, the linear feedback method extends the systemic dimension. The mathematical model of a four-dimensional structured chaotic system based on the Lorenz system is as follows

$$\begin{cases} \dot{x}(t) = a(y(t) - x(t)) \\ \dot{y}(t) = bx(t) + cy(t) - x(t)z(t) + w(t) \\ \dot{z}(t) = x(t)y(t) - dz(t) \\ \dot{w}(t) = -hx(t) \end{cases}$$

(8)

there are only two nonlinear terms in system (8), $a$, $b$, $c$, $d$, and $h$ are the control parameters, while the values $a = 35$, $b = 7$, $c = 12$, $d = 3$ and $h = 5$, and the system enters a chaotic state, the trajectories of chaotic attractors are shown in Figure 3.

In reality, there is often a time lag between the state variables of a dynamical system. The evolutionary trend of a system is related to the current state of the system and the state at a certain time in the past, thus evolving a time-delay dynamical system. We introduce delay time in the mathematical model of the four-dimensional Lorenz disturbance term $px(t - \tau)$ to form a new chaotic time-delay system.

$$\begin{cases} \dot{x}(t) = a(y(t) - x(t)) + px(t - \tau) \\ \dot{y}(t) = bx(t) + cy(t) - x(t)z(t) + w(t) \\ \dot{z}(t) = x(t)y(t) - dz(t) \\ \dot{w}(t) = -hx(t) \end{cases}$$

(9)

The variables $p$ and $\tau$ in Equation (9) represent the hysteresis control parameter and the hysteresis time of the time-delay system, respectively, which can be appropriately assigned to drive the newly constructed system into a chaotic state. The power spectrum of the trajectory $x(t)$ and sequence of chaotic attractors for this system are shown in Figure 4, which are obtained by picking $p = 3$, $\tau = 1.4$ ms and initial values $[-20; 0; 0; 0]$.

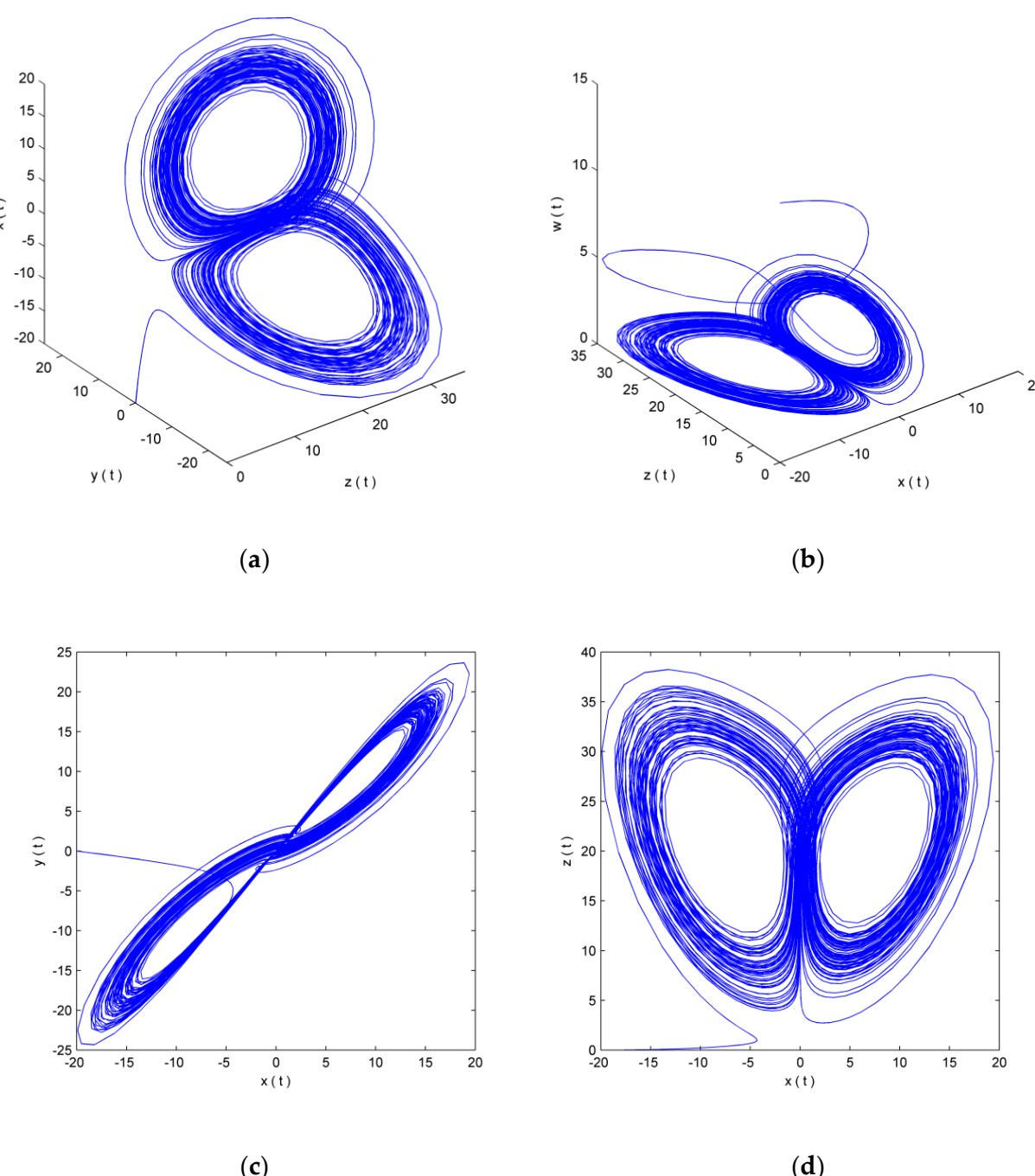

(**a**)

(**b**)

(**c**)

(**d**)

**Figure 3.** Phase diagram of a four-dimensional chaotic attractor. (**a**) *z-y-x* diagram; (**b**) *x-z-w* diagram; (**c**) *x-y* plane phase diagram; (**d**) *x-z* plane phase diagram.

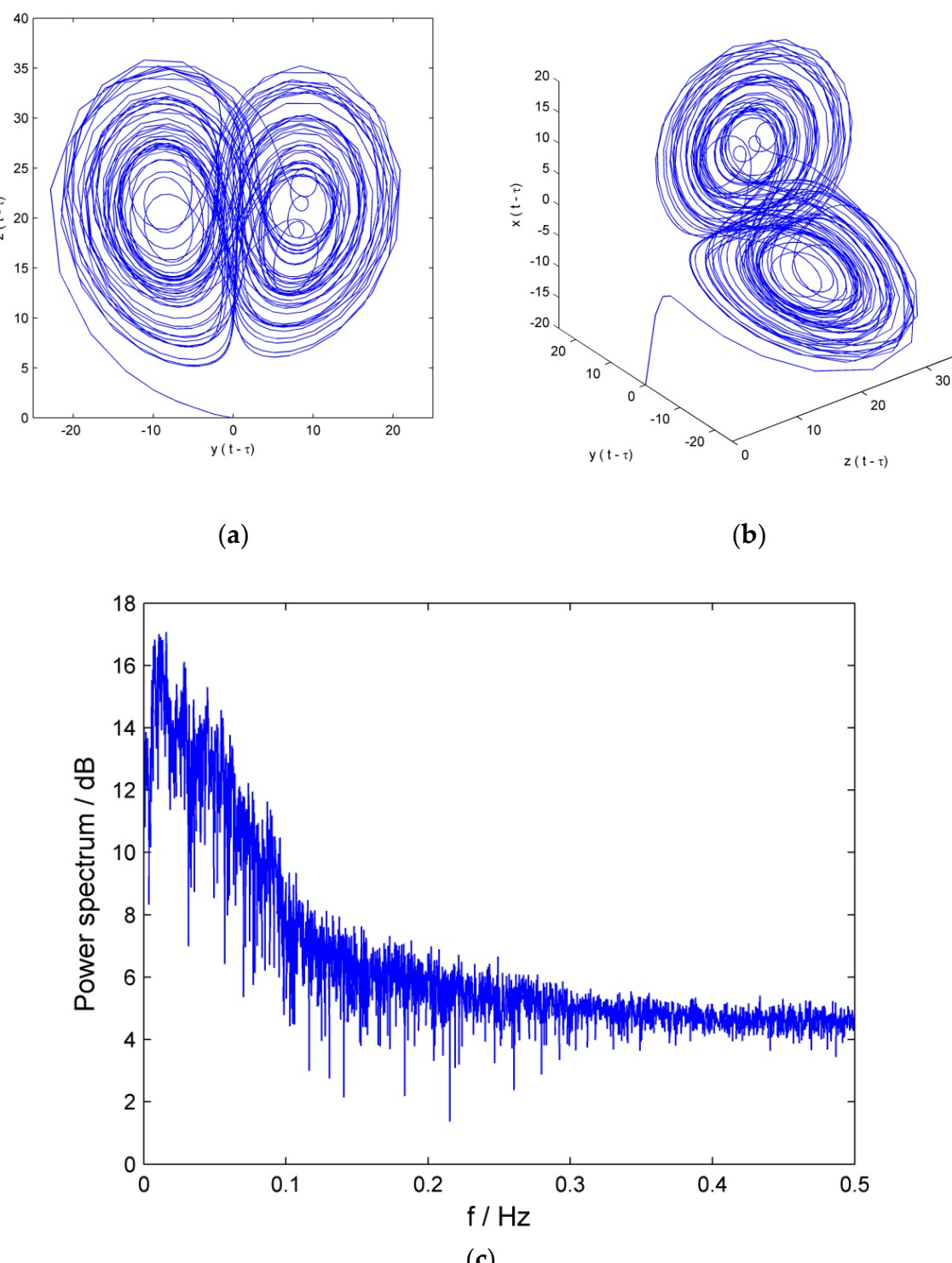

**Figure 4.** Phase diagram of a time-delay four-dimensional Lorenz chaotic attractor. (**a**) *x-z* plane phase diagram; (**b**) *x-y-z* plane phase diagram; (**c**) the power spectrum of the variable *x*(*t*).

The sensitivity of the initial values is one of the most important properties of chaos, which makes long-term prediction of the system impossible. Figure 5 shows the time domain waveform of the initial function $x(t)$ of the chaotic time-delay model combined with the difference between the $10^{-8}$ disturbance and the original series as a function of time $t$.

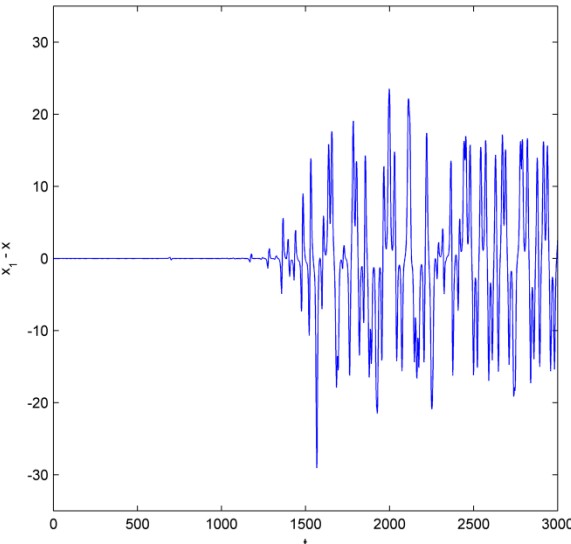

**Figure 5.** $x_1$-$x$ difference trajectory diagram.

The results show that the differences curve into a random vibration after some time. Its maximum difference can reach 31.7147, indicating that the trajectory of chaotic vibration is extremely sensitive to the initial value. The nonperiodic nature of the time domain waveform makes the system hard to predict. That is a consequence of positive Lyapunov exponents acting on the time-lag system (9). Figure 6 describes the Lyapunov exponential graph as it varies with the steps when the initial values are $[-20; 0; 0; 0]$, control parameters $a = 35$, $b = 7$, $c = 12$, $d = 3$, $h = 5$, $p = 3$, and delay-time $\tau = 1.4$ ms. There are two positive Lyapunov exponents when the evolution of the Lyapunov exponent is stable, respectively $\lambda_1 = 0.6607$, $\lambda_2 = 0.0524$.

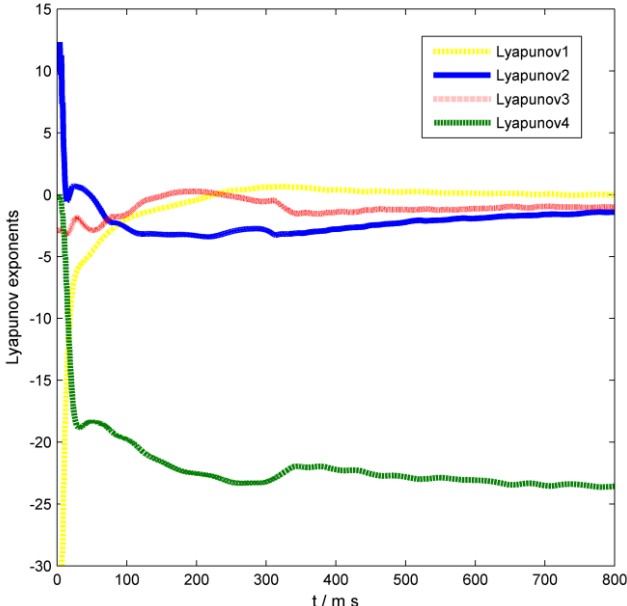

**Figure 6.** Lyapunov exponents of the time-delay system.

The maximum Lyapunov exponent of the chaotic attractor is measured to be 0.6607. Thus, the time-delay chaotic attractor constructed by introducing a perturbation term is more complex than the classical three-dimensional Lorenz attractor in dynamic behavior. This property is particularly valuable in secure communication.

Analyze the stability of its system equilibrium point

$$
\begin{cases}
a(y(t) - x(t)) + px(t - \tau) = 0 \\
bx(t) + cy(t) - x(t)z(t) + w(t) = 0 \\
x(t)y(t) - dz(t) = 0 \\
-hx(t) = 0
\end{cases}
\tag{10}
$$

At the equilibrium point, the solution of Equation (10) is a constant independent of time t. It is a constant state solution $x = x_0$, $y = y_0$, $z = z_0$, $w = w_0$. It cannot give a track line that changes with time, and there can be only a one-track line through each point on the phase plane except the equilibrium point, implying that the tracks cannot intersect. The solution gives the system only one equilibrium point $S_0 = (0, 0, 0, 0)$.

Stability analysis of this equilibrium point is performed by giving the system a small perturbation $\delta x, \delta y, \delta z, \delta w$ to move out of equilibrium. Assuming that the solution of the system (9) is $x = x_0 + \delta x$, $y = y_0 + \delta y$, $z = z_0 + \delta z$, $w = w_0 + \delta w$, and substituting them into the original system (9) and expanding the right-hand side of the equation to the linear term in Taylor's series form, we obtain at the equilibrium point

$$
\begin{cases}
\delta \dot{x} = f_1(x_0, y_0, z_0, w_0) + \frac{\partial f_1}{\partial x}\delta x + \frac{\partial f_1}{\partial y}\delta y + \frac{\partial f_1}{\partial z}\delta z + \frac{\partial f_1}{\partial w}\delta w \\
\delta \dot{y} = f_2(x_0, y_0, z_0, w_0) + \frac{\partial f_2}{\partial x}\delta x + \frac{\partial f_2}{\partial y}\delta y + \frac{\partial f_2}{\partial z}\delta z + \frac{\partial f_2}{\partial w}\delta w \\
\delta \dot{z} = f_3(x_0, y_0, z_0, w_0) + \frac{\partial f_3}{\partial x}\delta x + \frac{\partial f_3}{\partial y}\delta y + \frac{\partial f_3}{\partial z}\delta z + \frac{\partial f_3}{\partial w}\delta w \\
\delta \dot{w} = f_4(x_0, y_0, z_0, w_0) + \frac{\partial f_4}{\partial x}\delta x + \frac{\partial f_4}{\partial y}\delta y + \frac{\partial f_4}{\partial z}\delta z + \frac{\partial f_4}{\partial w}\delta w
\end{cases}
\tag{11}
$$

Equation (11) is a linear equation with a perturbation of $\delta x \delta y \delta z \delta w$. Jacobin matrix can be obtained at the right end, recorded as $J$

$$
J = \begin{pmatrix}
\frac{\partial f_1}{\partial x} & \frac{\partial f_1}{\partial y} & \frac{\partial f_1}{\partial z} & \frac{\partial f_1}{\partial w} \\
\frac{\partial f_2}{\partial x} & \frac{\partial f_2}{\partial y} & \frac{\partial f_2}{\partial z} & \frac{\partial f_2}{\partial w} \\
\frac{\partial f_3}{\partial x} & \frac{\partial f_3}{\partial y} & \frac{\partial f_3}{\partial z} & \frac{\partial f_3}{\partial w} \\
\frac{\partial f_4}{\partial x} & \frac{\partial f_4}{\partial y} & \frac{\partial f_4}{\partial z} & \frac{\partial f_4}{\partial w}
\end{pmatrix}
= \begin{pmatrix}
-a & a & 0 & 0 \\
b - z & c & -x & 1 \\
y & x & -d & 0 \\
-h & 0 & 0 & 0
\end{pmatrix}
= \begin{pmatrix}
-a & a & 0 & 0 \\
b & c & 0 & 1 \\
0 & 0 & -d & 0 \\
-h & 0 & 0 & 0
\end{pmatrix}
\tag{12}
$$

Linearizing the system (12) at the equilibrium point $S_0$ and constructing the characteristic equation as

$$
|J_s - \lambda I| = \left| \begin{pmatrix}
-a & a & 0 & 0 \\
b & c & 0 & 1 \\
0 & 0 & -d & 0 \\
-h & 0 & 0 & 0
\end{pmatrix} - \begin{pmatrix}
\lambda & 0 & 0 & 0 \\
0 & \lambda & 0 & 0 \\
0 & 0 & \lambda & 0 \\
0 & 0 & 0 & \lambda
\end{pmatrix} \right| = \begin{vmatrix}
-a - \lambda & a & 0 & 0 \\
b & c - \lambda & 0 & 1 \\
0 & 0 & -d - \lambda & 0 \\
-h & 0 & 0 & -\lambda
\end{vmatrix} = 0
\tag{13}
$$

from the determinant, we get

$$
\lambda^4 + (a - c + d)\lambda^3 + (ad - ac - ab - cd)\lambda^2 + (abd + ah - acd)\lambda + adh = 0
$$
$$
\lambda^4 + 26\lambda^3 + (-596)\lambda^2 + (-350)\lambda + 525 = 0
\tag{14}
$$

The sufficient conditions for the stability of a linear system are obtained by the Routh–Hurwitz criterion, which is an algebraic criterion for determining the stability of a system proposed by Routh in 1877, using the coefficients of the characteristic equation of a linear system as the criterion. Based on the coefficients of the characteristic Equation (14) and Routh–Hurwitz criterion Equation (6), the Routh table is presented in Table 1.

**Table 1.** System stability analysis.

| Eienvalue | $\Delta_1$ | $\Delta_2$ | $\Delta_3$ |
|---|---|---|---|
| $\lambda^4$ | 1 | $-596$ | 525 |
| $\lambda^3$ | 26 | $-350$ | 0 |
| $\lambda^2$ | $\frac{26\times(-596)-1\times(-350)}{26} \approx -582.5$ | $\frac{26\times525-1\times0}{26}=525$ | 0 |
| $\lambda^1$ | $\frac{-582.5\times(-350)-26\times525}{-582.5} \approx 373.4$ | 0 | 0 |
| $\lambda^0$ | 525 | 0 | 0 |

The linear system (14) is stable only if the values in the first column $\Delta_1$ of the Routh table are positive. Since the first column $\Delta_1$ of Table 1 has two variable signs, the system (14) is unstable. It has two positive real roots, proving that the system (9) is unstable at the equilibrium point $S_0$, which creates the possibility of a chaotic state.

System (9) is a dissipative system, and the dispersion is

$$\nabla V = \frac{\partial \dot{x}}{\partial x} + \frac{\partial \dot{y}}{\partial y} + \frac{\partial \dot{z}}{\partial z} + \frac{\partial \dot{w}}{\partial w} = -(a-c+d+h-p) = -28$$
$$V(t) = V(0)e^{-28} \tag{15}$$

When $(a-c+d+h-p) > 0$, $t \to \infty$, each volume element containing the system orbit contracts at an exponential rate of $-28$. The initial volume element $V(0)$ contracts at time t to the volume element $V(0)e^{-28}$ on an attractor.

## 3. Time-Delay Chaotic Synchronization Structure

In recent years, chaos has been used extensively in secure communication. Chaos synchronization has long been studied in the context of low-dimensional chaotic systems with only one positive Lyapunov exponent, and synchronized communication schemes using such low-dimensional chaos are easily deciphered by prediction and reconstruction methods. In contrast, high-dimensional hyperchaotic communication systems based on multiple positive Lyapunov exponents have better secrecy than low-dimensional chaotic systems. However, the structure is more complex, making chaos synchronization more difficult.

Time-delay chaotic systems are infinite-dimensional systems with more complex dynamical behavior, capable of generating positive Lyapunov exponents in more dimensions. The study of their synchronous systems is of great value in practical applications, making them a key focus in the study of synchronization of hyperchaotic systems [14].

The Lyapunov first and second methods are tools for studying the stability of general continuous dynamical systems described by ordinary differential equations. The first method determines the stability of a system based on the eigenvalues of differential equations. In contrast, the second method uses Lyapunov functions to analyze the global stability of the system. For time-delay chaos, the Lyapunov exponent of the system is related to a function on the initial period. Time-delay chaos is an infinite-dimensional system. It is more difficult to analyze the accuracy of the Lyapunov exponent if it is solved using local linearization. Using the Lyapunov first method in analyzing time-delay chaos is not advisable. Therefore, the extension of the Lyapunov second method to continuous dynamical systems with time lags in simultaneous method determination is called Lyapunov generalization.

Set $C[-r,0]$ to be the space of all continuous functions from $[-r,0]$ to $R^n$, and $r > 0$, for any $\phi \in C[-r,0]$, whose parameterization is $\|\phi\| = \sup_{-r \leq \theta \leq 0} \|\phi\|$, where $\|\cdot\|$ is the parameterization in $R^n$. When $x(t)$ is a continuous function on $[-r,T](0 \leq T \leq +\infty)$, define $x_t(\theta) = x(t+\theta)$, for any $t \in [0,T]$, $\theta$ takes all values on $[-r,0]$.

A general system of continuous differential dynamics with time delays is as follows

$$\dot{x} = F(t, x_t) \tag{16}$$

where $F : R \times C \to R^n$ is continuous, and $F(t, 0) = 0$, ensuring the existence and uniqueness of solution $x(t, t_0, \phi)$ for all initial value $(t_0, \phi)$, Equation (16), denoted $x(t) = x(t, t_0, \phi)$.

Let $V : R \times C \to R$ be a continuous function and define

$$\dot{V} = \dot{V}(t, \varphi) = \overline{\lim_{h \to 0^+}} \frac{1}{h} (V(t + h, x_{t+h}) - V(t, \phi)) \tag{17}$$

The Lyapunov functional method for stability of time-delay systems is given. Set $u(s), v(s), w(s) : R^+ \to R^+$ to be continuous and non-decreasing functions. When $s > 0$, $u(s) > 0, v(s) > 0, u(0) = v(0) = w(0) = 0$, then

1.  A zero solution $x = 0$ of Equation (16) is consistently stable if the function $V : R \times C \to R$ exists, such as, $u(\|\phi(0)\|) \leq V(t, \phi) \leq v(\|\phi\|), \dot{V}(t, \phi) \leq -w(\|\phi(0)\|)$.
2.  The zero solution of Equation (16) is uniformly bounded if the condition $\lim_{s \to +\infty} u(s) = +\infty$ is added to (1).
3.  If the condition $s > 0, w(s) > 0$ is added to (1), then the zero solution of Equation (16) is consistently asymptotically stable.

Previously, the time-lag phenomenon was often treated with a suppression approach in the design of chaotic synchronous systems, leading the system to an unstable periodic orbit. However, chaotic time-delay synchronous research has gradually developed to achieve complete reconfiguration of the chaotic states of two chaotic time-delay systems, which reduces the complexity and increases the reliability of the synchronized system [11].

Therefore, a multi-dimensional self-time-lagged chaotic synchronization method is found based on Lyapunov's general function theory, and the error system is verified through numerical simulations. Finally, the relevant factors affecting the convergence speed of the self-time-lagged chaotic synchronization are studied. The four-dimensional Lorenz system is taken as the driving source for the self-time-lag synchronous system, and the time-lag response system is as follows

$$\begin{cases} \dot{\widetilde{x}}(t) = a(\widetilde{y}(t) - \widetilde{x}(t)) + p\widetilde{x}(t - \tau) + u_1(t) \\ \dot{\widetilde{y}}(t) = b\widetilde{x}(t) + c\widetilde{y}(t) + \widetilde{w}(t) - \widetilde{x}(t)\widetilde{z}(t) + u_2(t) \\ \dot{\widetilde{z}}(t) = \widetilde{x}(t)\widetilde{y}(t) - d\widetilde{z}(t) + u_3(t) \\ \dot{\widetilde{w}}(t) = -h\widetilde{x}(t) + u_4(t) \end{cases} \tag{18}$$

It is crucial to design a suitable synchronous controller for the chaotic synchronization of the two systems. Based on the theory of active control, the choice of controllers $u_1(t), u_2(t), u_3(t), u_4(t)$ can achieve the global stability of the synchronized system.

$$\begin{cases} u_1(t) = -l_1 e_x(t) - px(t - \tau) \\ u_2(t) = -l_2 e_y(t) - x(t)z(t) + \widetilde{x}(t)\widetilde{z}(t) \\ u_3(t) = x(t)z(t) - \widetilde{x}(t)\widetilde{z}(t) - l_3 e_z(t) \\ u_4(t) = -l_4 e_w(t) \end{cases} \tag{19}$$

where $e_x = \widetilde{x} - x, e_y = \widetilde{y} - y, e_z = \widetilde{z} - z, e_w = \widetilde{w} - w$, the response system (18) is differenced from the four-dimensional Lorenz system to obtain the corresponding error system

$$\begin{cases} \dot{e}_x(t) = a(e_y(t) - e_x(t)) + p\widetilde{x}(t - \tau) + u_1(t) \\ \dot{e}_y(t) = be_x(t) + ce_y(t) + e_w(t) - \widetilde{x}(t)\widetilde{z}(t) + x(t)z(t) + u_2(t) \\ \dot{e}_z(t) = \widetilde{x}(t)\widetilde{z}(t) - x(t)z(t) - de_z(t) + u_3(t) \\ \dot{e}_w(t) = -he_x(t) + u_4(t) \end{cases} \tag{20}$$

$e_i(i = x, y, z, w)$ is called a chaotic synchronous error. According to the Lyapunov stability theorem, the synchronous error system (20) is asymptotically stable at the origin, which means the drive system (8) and the response system (18) are completely synchronized when the synchronous error tends to zero. The Lyapunov function is constructed

$$V(t) = \frac{1}{2}(e_x^2(t) + e_y^2(t) + e_z^2(t) + e_w^2(t)) + \int_{-\tau}^{0} (e_x^2(t+\theta) + e_y^2(t+\theta) + e_z^2(t+\theta) + e_w^2(t+\theta))\mathrm{d}\theta \tag{21}$$

for all $t \geq 0$, it is clearly shown that $V(t)$ is a positive-definite function. Evaluating the time derivative of $V(t)$ along the trajectory shown in the error system (4) gives

$$
\begin{aligned}
\dot{V}(t) &= e_x(t)\dot{e}_x(t) + e_y(t)\dot{e}_y(t) + e_z(t)\dot{e}_z(t) + e_z(t)\dot{e}_z(t) + [(e_x^2(t) + e_y^2(t) + \\
&\quad e_z^2(t) + e_w^2(t)) - (e_x^2(t-\tau) + e_y^2(t-\tau) + e_z^2(t-\tau) + e_w^2(t-\tau))] \\
&= -(l_1 + a - 1)e_x^2(t) - (l_2 - c - 1)e_y^2(t) - (l_3 + d - 1)e_z^2(t) - (l_4 - 1)e_w^2(t) \\
&\quad +(a+b)e_x(t)e_y(t) + e_y(t)e_w(t) - he_x(t)e_w(t) + pe_xe(t-\tau) \\
&\quad -(e_x^2(t-\tau) + e_y^2(t-\tau) + e_z^2(t-\tau) + e_w^2(t-\tau))]
\end{aligned}
\tag{22}
$$

according to arithmetic-geometric mean inequality

$$
\begin{aligned}
\dot{V}(t) &\leq -(l_1 + a - 1)e_x^2(t) - (l_2 - c - 1)e_y^2(t) - (l_3 + d - 1)e_z^2(t) - (l_4 - 1)e_w^2(t) \\
&\quad +(a+b)e_x(t)e_y(t) + e_y(t)e_w(t) - he_x(t)e_w(t) + \frac{p^2}{4}e_x^2 + e_x^2(t-\tau) \\
&\quad -(e_x^2(t-\tau) + e_y^2(t-\tau) + e_z^2(t-\tau) + e_w^2(t-\tau))] \\
&\leq -(l_1 + a - 1 - p^2/4)e_x^2(t) - (l_2 - c - 1)e_y^2(t) - (l_3 + d - 1)e_z^2(t) \\
&\quad -(l_4 - 1)e_w^2(t) + (a+b)e_x(t)e_y(t) + e_y(t)e_w(t) - he_x(t)e_w(t)
\end{aligned}
\tag{23}
$$

Assume $e^T(t) = (e_x, e_y, e_z, e_w)$, taking on the structure of the quadratic form

$$\dot{V}(t) \leq -e^T(t) \begin{pmatrix} l_1 + a - 1 - \frac{p^2}{4} & -a - b & 0 & h \\ 0 & l_2 - c - 1 & 0 & -1 \\ 0 & 0 & l_3 + d - 1 & 0 \\ 0 & 0 & 0 & l_4 - 1 \end{pmatrix} e(t) \tag{24}$$

By Krasovskii's sufficient condition, the error system (4) is asymptotically stable when the derived function $\dot{V}(t)$ is negative as the control parameters $l_1 > 1 - a + p^2/4$, $l_2 > c + 1$, $l_3 > 1 - d$, $l_4 > 1$, achieving the chaotic time-delay synchronization of the drive system and the response system. When delay time $\tau = 1.4$ms, control parameters $p = 3, l_1 = -1, l_2 = 24, l_3 = 2, l_4 = 4$, the initial values of the drive Equation (8) are $[-17; 7; 18; 15]$, the initial values of response Equation (18) are $[8; -9; 6; 20]$, and the synchronous error curves can be obtained in Figure 7.

Through the curves $e_x, e_y, e_z, e_w$, the error system converges to zero in a relatively short time under active control, which shows that the time-lag synchronous system can achieve chaotic synchronization rapidly with good robustness and stability. The effects of the control parameter $l$, the initial value of the drive response system, and the time lag $\tau$ on the speed t of synchronous convergence are given.

Table 2 indicates that the speed of convergence on the delay synchronous system is more sensitive to changes in the control parameters. In contrast, the initial value of the drive response system and time delay has less influence on the convergence rate, and convergence speed slows down significantly with the control parameter $l$ increase. For further research, it is expected that the proposed time-delay system state can be adaptively adjusted by improving the control algorithm to ensure that the system can work dynamically in an optimal state.

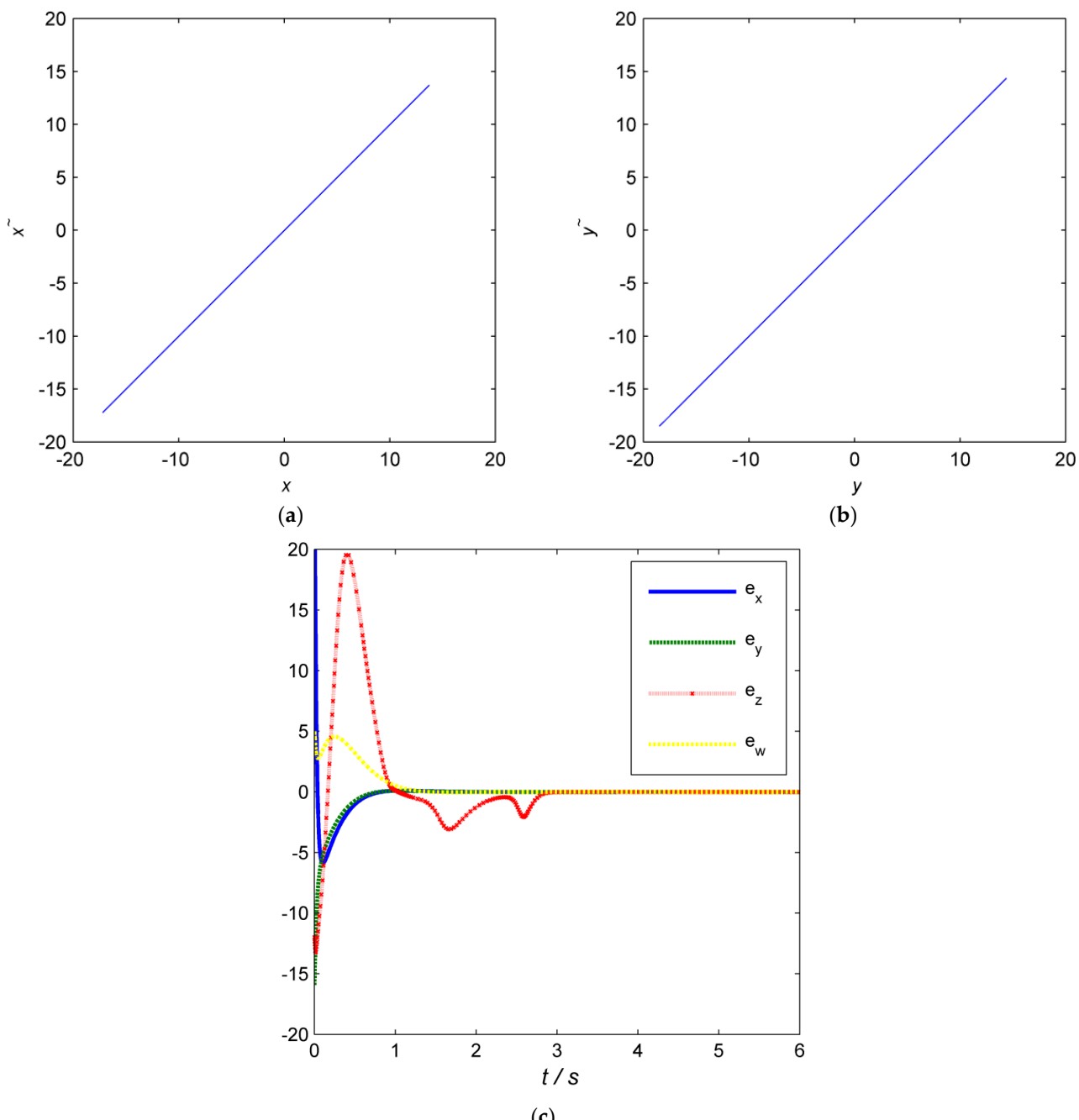

**Figure 7.** Diagram of the error synchronous process for time-delay systems. (**a**) Variable *x* curve; (**b**) variable *y* curve; (**c**) error curves.

**Table 2.** System synchronous convergence time comparison table.

| Initial values | [−19; 6; 20; 16; 10; −10; 8; 22] | | | | [−17; 7; 18; 15; 8; −9; 6; 20] | | | |
|---|---|---|---|---|---|---|---|---|
| *l* | [−3; 14; −1; 2] | | [−3; 14; −1; 2] | | [−1; 24; 2; 4] | | [−1; 24; 2; 4] | |
| $\tau/(\text{ms})$ | 1.4 | 2.4 | 1.4 | 2.4 | 1.4 | 2.4 | 1.4 | 2.4 |
| $t/(\text{s})$ | 12.5 | 12.7 | 11.9 | 10.5 | 14.7 | 14.8 | 13.5 | 14.0 |

#### 4. Time-Delay Chaotic Circuit

The growing research on chaos theory makes the connection between chaos and engineering even tighter. Purposeful enhancing and modeling of chaotic phenomena has become a pressing object of investigation [56]. In this section, we will define a detailed time-lag chaotic circuit with basic circuit components under the mathematical model of chaotic dynamics. Chaotic waveforms and chaotic phase diagrams comparable to the numerical study can be obtained from the oscilloscope. At the same time, the data were manipulated by computer, and various nonlinear dynamics parameters could be extracted, confirming that the chaotic system does coexist in nature.

We design simulation circuits for a four-dimensional chaotic time-delay system. The time-lag module is the central part of the system (9) simulation, which consists of a network of T-shaped LCL filters, as shown in Figure 8. As the reactance varies with frequency, the LC low-pass filter is connected to an inductor at serial and a capacitor at parallel. The signal frequency limits the network constructed by the low-pass filter, showing high input impedance and high output impedance when the frequency is increased. Test results showed that the time lag unit has smooth characteristics below the cutoff frequency 1 kHz.

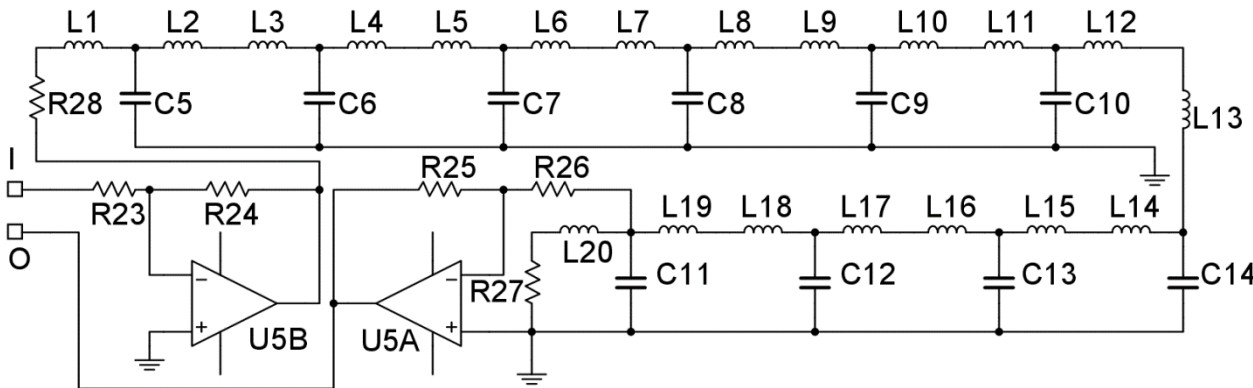

**Figure 8.** Time-delay module circuit.

A multi-stage filter is formed by placing $n = 10$ filter modules between the input and output ports to avoid single-stage filters failing when the useful signal frequency and the noise frequency are close together. Two matching resistors $R_{27}, R_{28} = 1\ \text{k}\Omega$ are placed at each port, and the characteristic impedance in the passband is constant. The time lag $\tau$ can be calculated by taking $R_{23} = R_{24} = R_{26} = 10\ \text{k}\Omega$, $R_{20} = R_{23} = 1\ \text{k}\Omega$ in the following equation

$$\tau = n\sqrt{2LC} \tag{25}$$

By taking $L = 20\ \text{mH}, C = 500\ \text{nF}, \tau = 1.4\ \text{ms}$, the calculated delay time $t$ remains the same as the value of delay time $t$ set by the time-delay chaotic system (9). After testing the simulation circuit, we found that the $V_x$ variable channel frequency is maintained at about 700 Hz, much smaller than the low-pass filter cutoff frequency $f_c$, thus ensuring that the signal passes through the module with low loss. To test the time-lag effect of the unit, we use a signal generator to input a 700 Hz sine wave to it and, comparing the input and output waveforms in Figure 9, we find that the output delay time is about 1.4 ms.

Multisim software is used to simulate the oscillator circuit of the time-delay system (2). The operational amplifier LF347BD and the analog multiplier used in the circuit are active devices. The typical working voltage of LF347BD is $\pm15$ V, and the linear dynamic range is only $\pm13.5$ V. It is necessary to ensure that the device voltage output value does not exceed its working voltage and linear dynamic range, otherwise they may appear saturated distortion, affecting the display effect of chaotic graphics. We linearly transform the system (9) so that its signal output level is 10% of the original, let

$$x = 10V_x\ , \ y = 10V_y\ , \ z = 10V_z\ , \ w = 10V_w \tag{26}$$

Equation (9) is adjusted as

$$\begin{cases} \dot{V}_x = a(V_y - V_x) - pV_x(\tau) \\ \dot{V}_y = bV_x + cV_y - 10V_xV_z + V_w \\ \dot{V}_z = 10V_xV_y - dV_z \\ \dot{V}_w = -hV_x \end{cases} \tag{27}$$

The circuit in Figure 10 is designed with five inter-coupled channels to perform the integration operations of the four system variables $x(t), y(t), z(t), w(t)$ and the time-lag functions of the state variable $x(t)$. An analog multiplier is used to implement the nonlinear terms in the system to avoid changing the initial nonlinear characteristics of the chaotic system, an operational amplifier is used to perform the addition and subtraction operations of the circuit, and linear resistors and capacitors are applied to assist in the addition, subtraction, multiplication and differentiation operations.

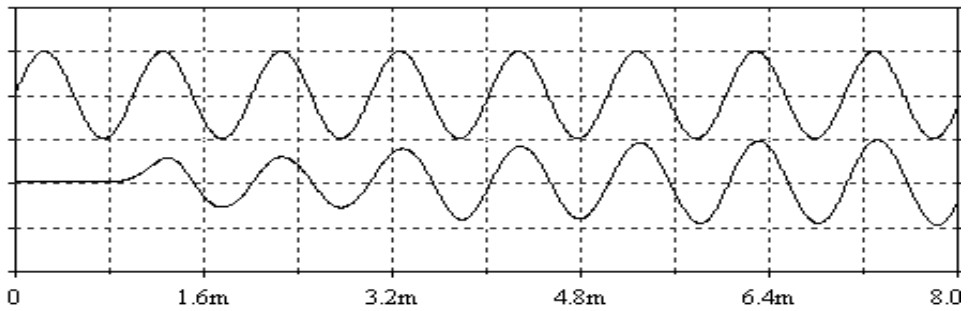

**Figure 9.** Time-delay waveform of sine signal (0.8 ms/div, 1 V/div).

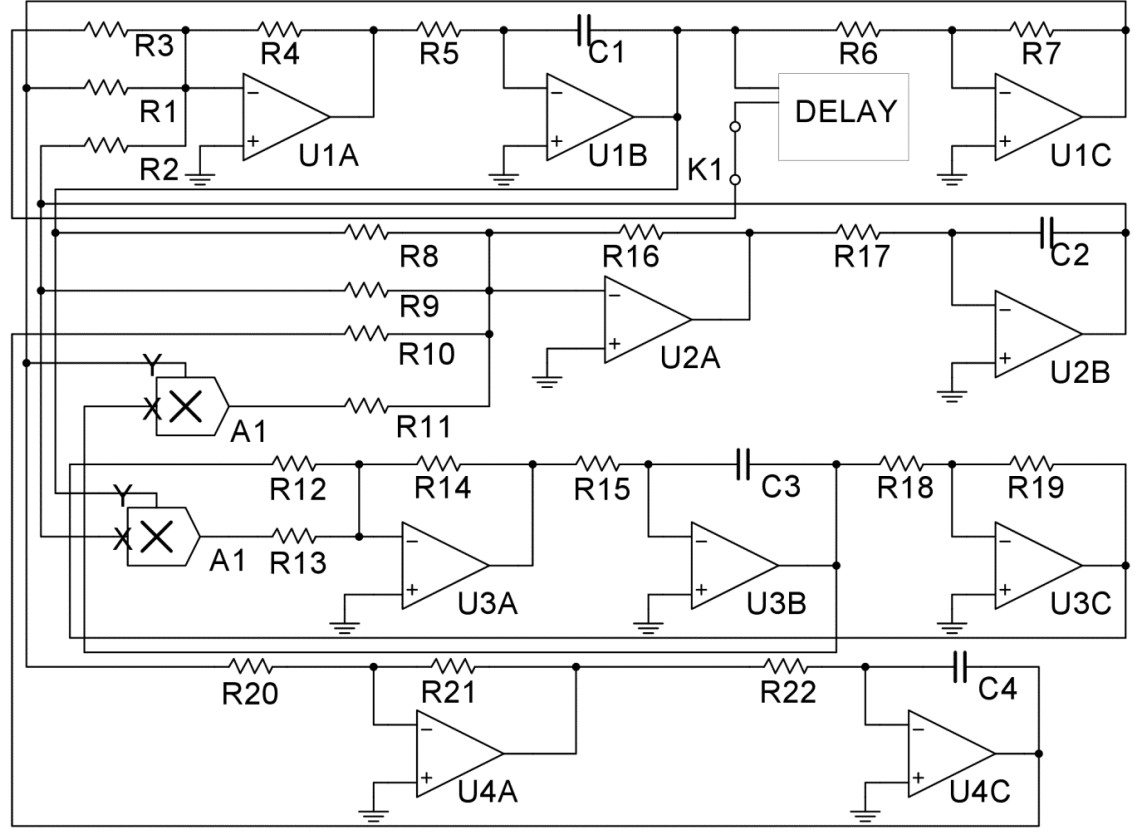

**Figure 10.** Time-delay Lorenz chaotic circuit.

Since no voltage is applied to the capacitor before integration, the initial value of its terminal voltage is zero, which is negligible. The output gains $G_1, G_2$ of the analog multiplier in the $y(t), z(t)$ variable channels are set to 0.1. According to the values of the system parameters, the resistance values are obtained as $R_1$, $R_2$, $R_6$, $R_7$, $R_{14}$, $R_{16}$, $R_{21}$ = 10 k$\Omega$, $R_5$, $R_{10}$, $R_{15}$, $R_{17}$, $R_{22}$ = 100 k$\Omega$, $R_{11}$, $R_{13}$ = 1 k$\Omega$, $R_4$ = 35k$\Omega$, $R_8$ = 15 k$\Omega$, $R_9$ = 8.3 k$\Omega$, $R_{20}$ = 20 k$\Omega$, and the integrator capacitance $C_1, C_2, C_3, C_4$ are 1 $\mu$F. Based on the nodal voltage method to analyze the structure shown in Figure 10 and substituting the parameters into Equation (28), it can be verified that the mathematical model of the circuit is the same as the system (9).

$$\begin{cases} \dot{V}_x = -\frac{R_4 R_7}{R_1 R_5 R_6 C_1} V_x + \frac{R_4}{R_2 R_5 C_1} V_y + \frac{R_4}{R_3 R_5 C_1} V_x(\tau) \\ \dot{V}_y = \frac{R_{16}}{R_8 R_{17} C_2} V_x + \frac{R_{16}}{R_9 R_{17} C_2} V_y - \frac{G_1 R_7 R_{16}}{R_6 R_{11} R_{17} C_2} V_x V_z + \frac{R_{16}}{R_{10} R_{17} C_2} V_w \\ \dot{V}_z = -\frac{R_{14} R_{19}}{R_{12} R_{15} R_{18} C_3} V_z + \frac{G_2 R_{14}}{R_{13} R_{15} C_3} V_x V_y \\ \dot{V}_w = -\frac{R_7 R_{21}}{R_6 R_{20} R_{22} C_4} V_x \end{cases} \tag{28}$$

We reduce the integral capacitances $C_1, C_2, C_3, C_4$ to 1 nF to adjust the capacitance multiplier, and thus can avoid interference from high-frequency signals, which means increasing the output signal frequency by a factor of 1000 while maintaining the original system properties, changing only the time series of the signal.

The experiment was conducted in Multisim, and the results were presented in Figure 11. The circuit simulation of the time-lagged Lorenz system is consistent with the numerical analysis of the system (9), which proves the effectiveness of the method for high-dimensional time-delay chaotic systems and the feasibility of time-delay Lorenz circuits and lays an experimental foundation for the study of time-delay chaos in secure communication and practical engineering applications.

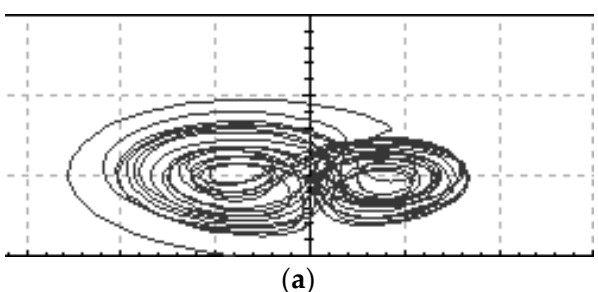
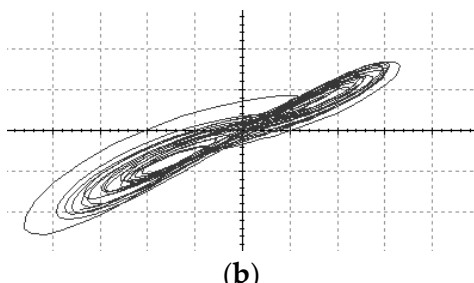

|(a)|(b)|

**Figure 11.** Two-dimensional planar projection of chaotic time-delay system. (**a**) $V_y$-$V_z$ (0.5 V/div, 1 V/div); (**b**) $V_x$-$V_y$ (1 V/div, 2 V/div).

## 5. Simulation of a Self-Synchronous Circuit for a Four-Dimensional Time-Delay Lorenz System

Researchers have gradually realized the value of chaos synchronization and its application in chaos theory research in recent years, such as adaptive, self-activating control, and backstepping methods. The coupled synchronous control method was used in the research of L.M. Pecora and T.L. Carroll in 1990. Then Chua conducted an in-depth analysis and study of the coupled synchronization of Chua's circuits, theoretically proving that as long as the coupling between two chaotic systems is strong enough, chaotic synchronization can be achieved. However, most of the coupled synchronous schemes that have been proposed are limited to chaotic systems with only one positive Lyapunov exponent. In contrast, relatively few studies have been carried out to realize coupled synchronization of multi-dimensional chaotic systems or even hyperchaotic systems. Extending the coupled synchronous scheme to hyperchaotic synchronous systems would be a research trend [14].

There are many basic methods for determining the existence of chaotic synchronization, such as the Routh–Hurwitz stability discrimination method, the method for solving the optimal solution of linear inequalities based on Lyapunov stability, and the Lyapunov

function method [56]. We will choose the Lyapunov function method to determine the existence of a coupled synchronous system model.

The dynamical equations of the chaotic system take the form

$$\dot{X}(t) = AX(t) + f(X(t), t) \tag{29}$$

where $X \in R^n$ is the n-dimensional state vector of the system, $A$ is the n-dimensional constant matrix, and $f$ is a smooth nonlinear continuous function, and adding the time-delay linear perturbation term gives

$$\dot{X}(t) = AX(t) + f(X(t), t) + PX(t - \tau) \tag{30}$$

$P$ is an n-dimensional constant matrix and $\tau$ is the time lag constant of the system. By choosing an appropriate time delay $\tau$ and linear perturbation $P$, the system (30) can be left in a chaotic state.

We take the system (29) as the driving system and obtain the response system as

$$\dot{\widetilde{X}}(t) = A\widetilde{X}(t) + f(\widetilde{X}(t), t) + P\widetilde{X}(t - \tau) + U(t) \tag{31}$$

where $U(t) = (U_1(t), U_2(t) \cdots U_n(t))$ is the synchronous controller.

For an initial value, the drive system (29) is said to have achieved synchronization with the response system (31) if it is such that

$$\lim_{t \leftarrow \infty} \|X(t) - \widetilde{X}(t)\| = 0 \tag{32}$$

the following error system is obtained by subtracting the general chaotic system (29) from the constructed time-lag chaotic system (31)

$$\dot{e}(t) = \dot{X}(t) - \dot{\widetilde{X}}(t) = Ae(t) - P\widetilde{X}(t - \tau) - f(\widetilde{X}(t), t) + f(X(t), t) - Qe(t) \tag{33}$$

The response system (31) can be adjusted without affecting the drive system (29) to achieve a coupled synchronization of the drive system (31) and the response system (33) with a suitably selected control gain $Q$.

From the median theorem, it follows that

$$f(X(t), t) - f(\widetilde{X}(t), t) = Ke(t) \tag{34}$$

where $K$ is the Jacobi matrix of $f$.

In Equation (33), the control matrix $Q = diag(q_1, q_2 \cdots q_n)$ is referred to as the feedback gain matrix

$$\dot{e}(t) = (A + K - Q)e(t) - P\widetilde{X}(t - \tau) = A_f(t)e(t) - P\widetilde{X}(t - \tau) \tag{35}$$

At this point, Equation (35) is a time-varying system of chi-square linear equations. An appropriate feedback gain matrix $Q$ can be chosen so that the eigenvalues of the matrix $A_f(t)$ all have negative real parts, allowing equation (32) to be implemented. The drive system (29) and response system (31) will be coupled and synchronized. Then, based on the Lyapunov stability theory, the analysis of the Lyapunov functional of the chaotic dynamical error system (35) yields

$$V(e(t)) = \frac{1}{2}e^T(t)e(t) + \int_{-\tau}^{0} e^T(t + \theta)e(t + \theta)d\theta \tag{36}$$

then the derivative of $V$ for time $t$ is

$$\dot{V}(e(t)) = e^T(t)\dot{e}(t) + e^T(t)e(t) - e^T(t-\tau)e(t-\tau)$$
$$= e^T(t)L_1e(t) - e^T(t-\tau)L_2e(t-\tau) \tag{37}$$

where $L_1, L_2$ are positive definite matrices, both represent the coefficient matrices after Lyapunov functional analysis.

By selecting the feedback gain matrix $Q$ appropriately, it is possible to make the system matrix $L_1$ negative and thus keep the derivative of $V(e(t))$ negative. In this case, the error system (33) is globally asymptotically stable near the origin, meaning that for any initial condition, $\lim\limits_{t\to\infty}\|e(t)\| = 0$.

We introduce the coupled synchronous scheme into a four-dimensional time-delay Lorenz chaotic system to achieve coupled chaotic synchronization of two time-lagged hyperchaotic systems with the same initial values, prove the corresponding asymptotic conditions for synchronous convergence, and innovatively propose a coupled synchronous control circuit that can be verified by simulation to demonstrate that the method is effective.

The proposed four-dimensional Lorenz system (8) is used as the driving system for the coupled synchronization and is written in matrix form as follows

$$\dot{X}(t) = AX(t) + f(X(t), t) \tag{38}$$

where $X(t) = [X_1(t), X_2(t), \cdots, X_4(t)]^T$,

$$A = \begin{bmatrix} -a & a & 0 & 0 \\ b & c & 0 & 1 \\ 0 & 0 & -d & 0 \\ -h & 0 & 0 & 0 \end{bmatrix}, \ f(X(t), t) = \begin{bmatrix} 0 \\ -x(t)z(t) \\ x(t)y(t) \\ 0 \end{bmatrix}$$

the constructed time-delay chaotic system (9) is taken as a coupled synchronous response system as follows

$$\dot{\widetilde{X}}(t) = A\widetilde{X}(t) + f(\widetilde{X}(t), t) + P\widetilde{X}(t-\tau) + Q(X(t) - \widetilde{X}(t)) \tag{39}$$

let us denote the error system

$$e(t) = X(t) - \widetilde{X}(t) = \begin{bmatrix} e_1(t) \\ e_2(t) \\ e_3(t) \\ e_4(t) \end{bmatrix}, \ e_i(t) = X_i(t) - \widetilde{X}_i(t)$$

the error system of the coupled system (39) and system (38) is

$$\dot{e}(t) = Ae(t) - P\widetilde{X}(t-\tau) - f(\widetilde{X}(t), t) + f(X(t), t) - Qe(t) \tag{40}$$

where

$$f(X(t), t) - f(\widetilde{X}(t), t) = \begin{bmatrix} 0 \\ -X(t)Z(t) + \widetilde{X}(t)\widetilde{Z}(t) \\ X(t)Y(t) - \widetilde{X}(t)\widetilde{Y}(t) \\ 0 \end{bmatrix}$$

$$= \begin{bmatrix} 0 & 0 & 0 & 0 \\ -Z(t) & 0 & -\widetilde{X}(t) & 0 \\ Y(t) & \widetilde{X}(t) & 0 & 0 \\ 0 & 0 & 0 & 0 \end{bmatrix} \bullet \begin{bmatrix} e_1(t) \\ e_2(t) \\ e_3(t) \\ e_4(t) \end{bmatrix} = Ke(t) \tag{41}$$

The synchronization of the coupled system (39) with the system (38) is achieved by simply satisfying certain conditions on the parameters of the coupling function so that its

error system is asymptotically stable as time tends to infinity, allowing the nonlinear coupling and synchronization of two uniform chaotic systems with different initial conditions but the same structure.

In the coupled system (39), the matrix of functions of the state variables is taken

$$Q = \begin{bmatrix} q_1 & -Z(t) & Y(t) & 0 \\ 0 & q_2 & 0 & 0 \\ 0 & 0 & q_3 & 0 \\ 0 & 0 & 0 & q_4 \end{bmatrix} \tag{42}$$

where $q_i \geq 0$ ($i = 1, 2, 3, 4$) are the parameters to be determined. We obtain the error system (40) as

$$\dot{e}(t) = Ae(t) + (K - Q)e(t) - P\widetilde{X}(t - \tau) \tag{43}$$

by calculating, we get

$$K - Q = \begin{bmatrix} -q_1 & Z(t) & -Y(t) & 0 \\ -Z(t) & -q_2 & -\widetilde{X}(t) & 0 \\ Y(t) & \widetilde{X}(t) & -q_3 & 0 \\ 0 & 0 & 0 & -q_4 \end{bmatrix} \tag{44}$$

then we can write

$$\frac{1}{2}\left[(K - Q) + (K - Q)^T\right] = diag(-q_1, -q_2, -q_3, -q_4) \tag{45}$$

We construct the Lyapunov function $V(e) = \frac{1}{2}e^T(t)e(t) + \int_{-\tau}^{0}(e^T(t+\theta)e(t+\theta))\mathrm{d}\theta$ and find its derivative for time $t$ as

$$\begin{aligned}
\dot{V}(e) &= e^T(t)\dot{e}(t) + [e^2(t) - e^2(t - \tau)] \\
&= e^T(t)Ae(t) + e^T(t)(K - Q)e(t) + [e^2(t) - e^2(t - \tau)] \\
&= e^T(t)\frac{A + A^T}{2}e(t) + e^T(t)Qe(t) + [e^2(t) - e^2(t - \tau)] \\
&= e^T(t)\mathrm{R}e(t) + [e^2(t) - e^2(t - \tau)]
\end{aligned} \tag{46}$$

the calculation yields

$$R = \begin{bmatrix} -a - q_1 & \frac{a+b}{2} & 0 & -\frac{h}{2} \\ \frac{a+b}{2} & c - q_2 & 0 & 0 \\ 0 & 0 & -d - q_3 & 0 \\ -\frac{h}{2} & 0 & 0 & -q_4 \end{bmatrix} \tag{47}$$

To make $R$ negative, we must meet the following conditions for

$$\begin{aligned}
(1)\ &\Delta_1 = -a - q_1 < 0 \\
(2)\ &\Delta_2 = (a + q_1)(q_2 - c) - \frac{(a+b)^2}{4} > 0 \\
(3)\ &\Delta_3 = -(d + q_3)\Delta_2 < 0 \\
(4)\ &\Delta_4 = -q_4\Delta_3 + \frac{h^2}{4}(c - q_2)(-d - q_3) > 0
\end{aligned} \tag{48}$$

We compute the principal subdivisions of the matrix $R$ at each level of order and know that

$$q_1 > -35, q_2 > 453, q_3 > -3, q_4 > 0, \tag{49}$$

At this point, $R$ is negative, thus the derivative of $V(e(t))$ is also negative, so the error system (42) is globally asymptotically stable at the origin, $\lim\limits_{t\to\infty}\|e(t)\| = 0$ for any initial condition.

Based on the chaotic time-delay system in the simulation experiment in Figure 10, we used Multisim software to design the univariate coupled time-lag chaotic oscillator

synchronous circuit shown in Figure 12. The solution involves controlling the synchronization of two constructive time-delay chaotic systems with similar evolutionary laws by appropriately driving the system variable $Y$ back to the response system, thus controlling the system to synchronize its variable coupled system.

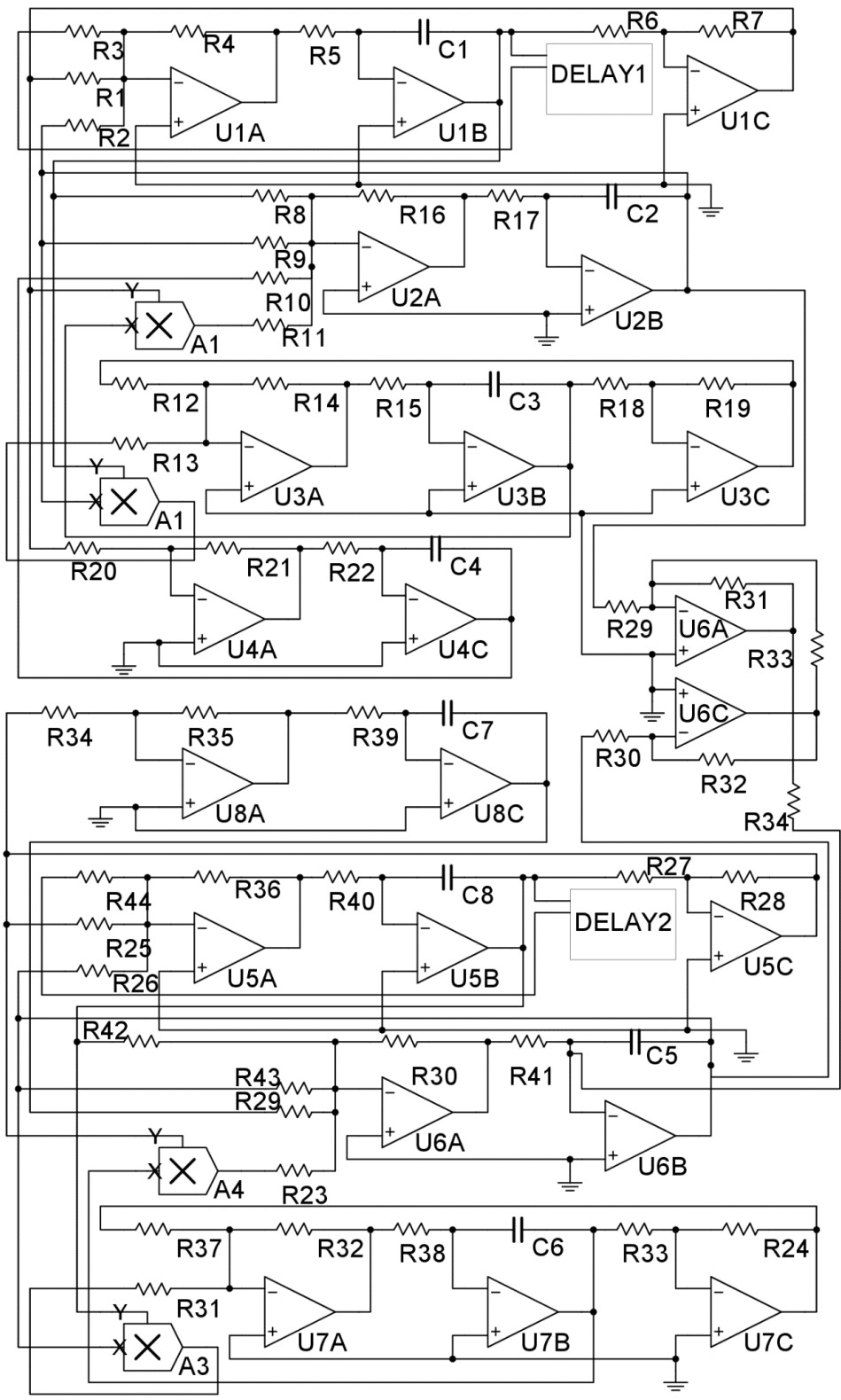

**Figure 12.** Constructed chaotic time-delay synchronous circuit.

We choose the chaotic system (8) as the driving system so that the response (controlled) system is

$$\begin{cases} \dot{x}_1(t) = a(y_1(t) - x_1(t)) + px_1(t - \tau) \\ \dot{y}_1(t) = bx_1(t) + cy_1(t) - x_1(t)z_1(t) + w_1(t) + q(y(t) - y_1(t)) \\ \dot{z}_1(t) = x_1(t)y_1(t) - dz_1(t) \\ \dot{w}_1(t) = -hx_1(t) \end{cases} \quad (50)$$

By choosing the appropriate control gain $q$, we can adjust the response system (50) without affecting the drive system (8) so that the state of the system (50) converges to the system (8) and is eventually fully synchronized.

We show the synchronous control circuit in Figure 13 with the variable $Y$ of the drive system (8) and the variable $Y1$ of the response system (50) at the two inputs, and the controlled feedback value $Y - Y1$ at the output. To illustrate the synchronous effect of this control circuit, we can use the nodal voltage method to derive the state equation $\widetilde{Y} - \widetilde{Y}1$ in front of the resistor $R_{34}$ as follows

$$V_{\widetilde{Y}-\widetilde{Y}1} = -\frac{R_{31}}{R_{29}}Y + \frac{R_{31}R_{32}}{R_{30}R_{33}}Y_1 \quad (51)$$

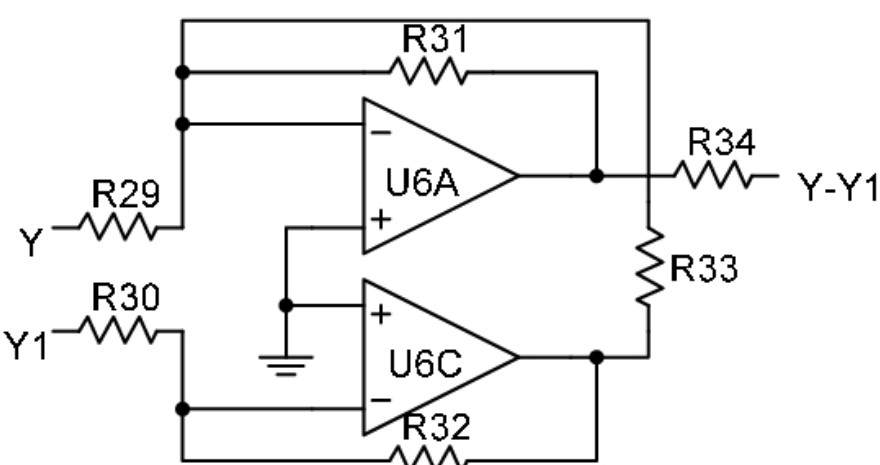

**Figure 13.** Synchronous control circuit.

Different control gain $q$ can be produced by tuning the resistance values of $R_{29}$, $R_{30}$, $R_{31}$, $R_{32}$, and $R_{34}$, as shown in Table 3.

**Table 3.** Synchronous control gain.

| $R_{29}, R_{30}, R_{31}, R_{32}$ (k$\Omega$) | [10, 10, 10, 10] | | | |
|---|---|---|---|---|
| $R_{34}$ (k$\Omega$) | 100 | 300 | 500 | 700 |
| control gain $q$ | 10 | 3.3 | 2 | 1.4 |

We compare the synchronous effect of this synchronous control circuit for different control gains and observe the system variable $Y - Y1$ on an oscilloscope. The synchronous phase diagrams are presented in Figure 14.

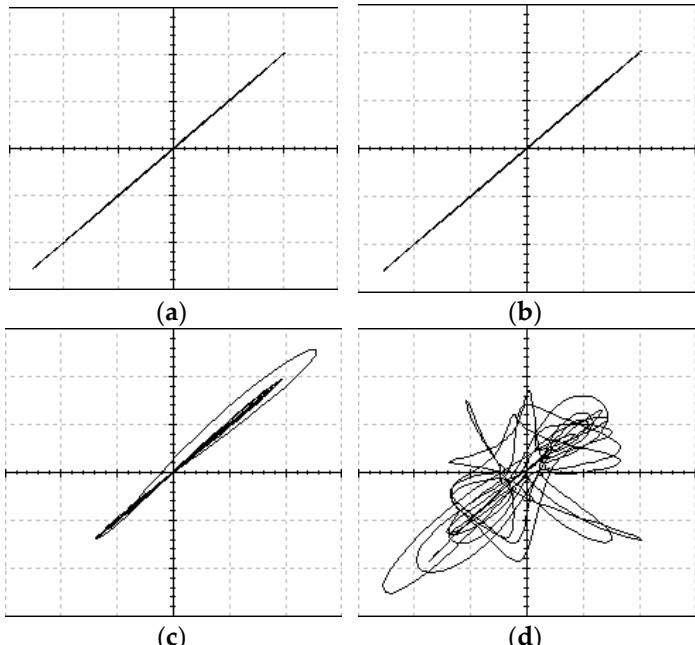

**Figure 14.** Synchronous effects with different control gains. (**a**) $q$ = 10 (1 V/div, 1 V/div); (**b**) $q$ = 3.3 (1 V/div, 1 V/div); (**c**) $q$ = 2 (1 V/div, 1 V/div); (**d**) $q$ = 1.4 (1 V/div, 1 V/div).

Through numerical simulations, we have found that the two time-lagged chaotic systems can only be synchronized gradually when the control gain $q > 2.5$, and that the chaos synchronous effect is proportional to the value of the gain.

The value of the gain $q$ can be adjusted independently by changing the value of the resistor $R_{34}$.

$$q = \frac{1}{R_{34} \cdot C} \tag{52}$$

The threshold value corresponding to the control gain is taken as $R_{34}$ = 400 k$\Omega$. When the gain is applied $q = 10$, a comparison of the waveforms of the system variable $Y - Y1$ is observed on an oscilloscope, as in Figure 15.

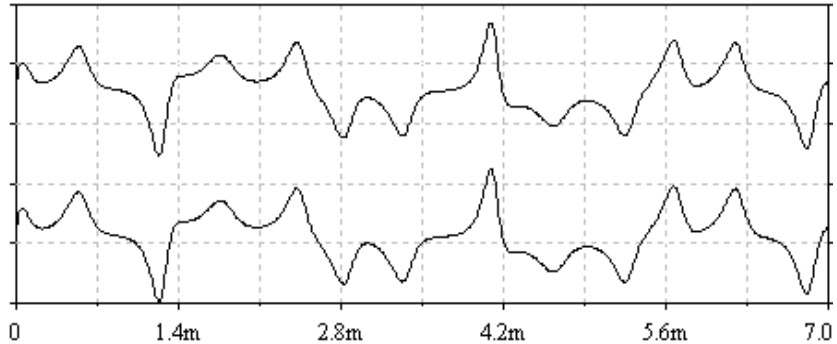

**Figure 15.** Comparison of synchronous signal waveforms (0.7 ms/div, 1 V/div).

Our synchronous experiments find that the univariate coupling synchronous control method did not work well enough for the other three coupled states $X - X1$, $Z - Z1$, and $W - W1$. Therefore, the system is most easily controlled and most effective when the input signal $Y - Y1$ is used for synchronous control. At this point, both the drive and response systems are controlled by each other's coupling, prompting the error between the drive and response systems to rapidly converge to the origin position, realizing a nonlinear coupled global asymptotic synchronization of the hyperchaotic system.

## 6. Conclusions

In this paper, we propose a new time-delay chaotic mathematical model with a simple topology, which is highly sensitive to initial values and can achieve stable synchronization in a short time. The sufficient conditions for synchronous convergence are derived through the construction of time-lag chaotic synchronization. The control parameter *l* influences the rate of synchronous convergence significantly. We have innovated the design of an oscillator simulation circuit for a time-delay system, applied the univariate coupled control method to achieve synchronous control on two four-dimensional time-delay chaotic systems, and gave a complete synchronous experimental circuit. These lay the foundation for the in-depth study of time-delay chaotic synchronous theory in secure and spread spectrum communication.

**Author Contributions:** Conceptualization, Z.C.; Data curation, Z.C.; Funding acquisition, Z.C. and X.Q.; Resources, D.Z.; Software, D.Z.; Supervision, X.Q.; Writing—original draft, Z.C.; Writing—review & editing, D.Z. and X.Q. All authors have read and agreed to the published version of the manuscript.

**Funding:** This research has been supported by the Jiangxi Natural Science Foundation (NO. 20181BAB202018), and the foundation of Jiangxi University of Science and Technology (Nos. XQZG150303 and XJG202026).

**Institutional Review Board Statement:** Not applicable.

**Informed Consent Statement:** Not applicable.

**Data Availability Statement:** Not applicable.

**Conflicts of Interest:** The authors declare no conflict of interest.

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
