# Peer review of "Synchronization Analysis of a New Four-Dimensional Time-Delay Lorenz System and Its Circuit Experiments"

_applsci, doi:10.3390/app122010557_

Round 1

Reviewer 1 Report

In line 82, a space is missing.

It is not clear the main contribution of the manuscript.

In organization of the paper, lines 134 and 137, Section 5 is named two times.

In Fig. 6, Lyapunov exponents are ploted vs time??? It would be better to mention that the three positive Lyapunov exponents do not occur at once; only two of them and for some values of the parameter in question.

In Eq. 19, where in the system is the delay included?, as authors say in line 386.

Line 381: Eq 22 is not a “positive symmetric matrix”. Then, the analysis is not correct.

In Eq. 22, in position a11, where does the term p2/4 come from?  In position a34, there is a “c”; Shouldn't it be "0"? What would be the form of the Lyapunov equation that delivers that derivative?

Control strategy needs all state from the master system. It is a disadvantage for any synchronization purpose.

Reviewer 2 Report

1. It is tough to get the ideas of many paragraphs because of poor English. 

2. In many places terms are used inappropriately- e.g. "chaotic state along with the presence of attractors...", "synchronous" vs "synchronization" etc. 

3. In some places the sizes of fonts are mixed. 

4. The quality of some Figures (e.g. 8, 10) should be improved.

5. Due to the poor language it is hard to understand all mathematical derivations.

6. It is also hard to get the Schematic in Fig. 12 comes from. 

Round 2

Reviewer 1 Report

Authors have attended all suggestions and they have clarified any question.

Reviewer 2 Report

Could be published.